# Influence of Movement Energy and Affect Priming on the Perception of Virtual Characters Extroversion and Mood

TANJA SCHNEEBERGER, FATIMA AYMAN ALY, DAKSITHA WITHANAGE DON, KATHARINA GIES, ZITA ZEIMER, FABRIZIO NUNNARI, PATRICK GEBHARD, German Research Center for Artificial Intelligence (DFKI), Saarland Informatics Campus D3 2, Saarbrücken, Germany

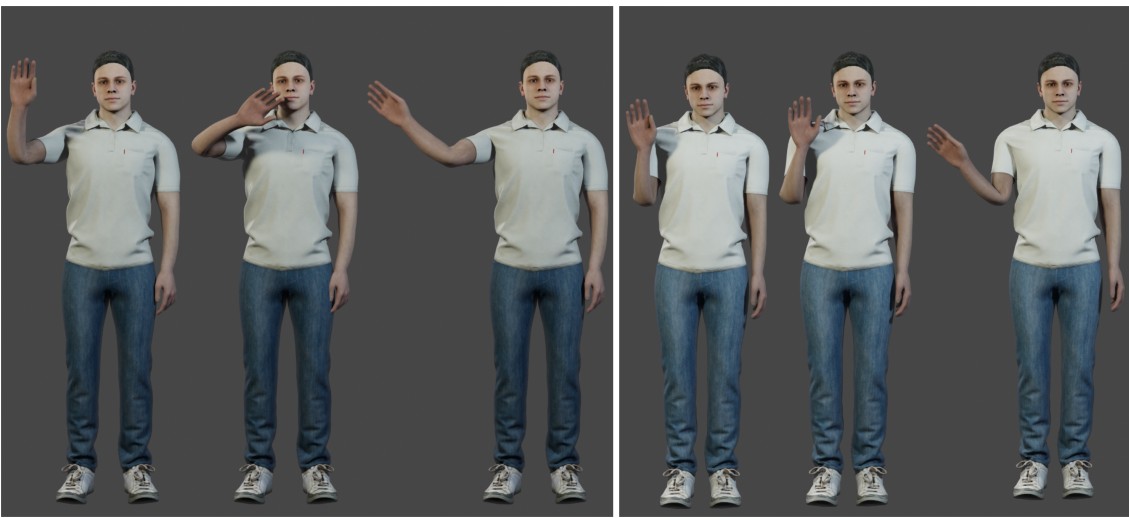

Fig. 1. Variants of male waving character derived from a neutral waving motion: (left) High energy movement, and (right) Low energy movement

Movement Energy – physical activeness in performing actions and Affect Priming – prior exposure to information about someone's mood and personality might be two crucial factors that influence how we perceive someone. It is unclear if these factors influence the perception of virtual characters in a way that is similar to what is observed during in-person interactions. This paper presents different configurations of Movement Energy for virtual characters and two studies about how these influence the perception of the characters' personality, extroversion in particular, and mood. Moreover, the studies investigate how Affect Priming (Personality and Mood), as one form of contextual priming, influences this perception. The results indicate that characters with high Movement Energy are perceived as more extrovert and in a better mood, which corroborates existing research. Moreover, the results indicate that Personality and Mood Priming influence perception in different ways. Characters that were primed as being in a positive mood are perceived as more extrovert, whereas characters that were primed as being introverted are perceived in a more positive mood.

CCS Concepts: • **Computing methodologies** → **Computer graphics**; Graphics systems and interfaces; Perception; *Procedural animation*; • **Human-centered computing** → **Empirical studies in HCI**;

Additional Key Words and Phrases: virtual characters, character animation, contextual priming, perceptual study

**ACM Reference Format:**
Tanja Schneeberger, Fatima Ayman Aly, Daksitha Withanage Don, Katharina Gies, Zita Zeimer, Fabrizio Nunnari, Patrick Gebhard. 2021. Influence of Movement Energy and Affect Priming on the Perception of Virtual Characters Extroversion and Mood. In *Companion Publication of the 2021 International Conference on Multimodal Interaction (ICMI '20 Companion), Oct 18–22, 2021, 2021, Montreal, Canada.* ACM, New York, NY, USA, 16 pages. https://doi.org/xxx

## 1 INTRODUCTION

That people have expectations towards computers is a well-known phenomenon [28]. This is even more pronounced when human-like cues in the computer interface occur [22]. Virtual characters (VC) with a human-like appearance have been shown to evoke communication behavior, and emotional reactions [28] that are equivalent to what would be expected in a human face-to-face conversation. In interactions with VCs and as observers, people seem to have expectations, like the personality of a VC. These expectations can have an effect on the perception of the VC, similar to the perception of humans. Expectations and perception might be influenced by VC's moving behavior [6]. There are computational models aiming to integrate personality as long-term affect and moods as medium-term affect into the behavior and motion of VCs [13, 33]. Though user's responses to VCs seem to be influenced by the reflection of affective states in their body motions [8], studies examining how humans perceive the interplay between personality and mood are rare.

Also, prior information can influence expectations towards a person or how it is perceived [26]. This contextual priming can have different sources, for example, the person's affect. "She is in a happy mood today" or "She has a rather introvert personality"; having this information about someone influences how people perceive others in an upcoming interaction. If the perception of a VC can be influenced by affect priming remains unknown. However, this knowledge might be crucial when designing applications in which VCs have to communicate different affective states.

In this paper, we investigate how 1) *Movement Energy* of VCs' motion and 2) *Affect Priming* influence the perception of VCs' personality and mood focusing on extroversion and introversion for the former as well as positive and negative for the latter. Movement Energy describes motion aspects such as speed, acceleration, position, and extension. Affect Priming describes contextual priming with affective information about a character. In this context, we first present how we animate a female and a male VC expressing extrovert and introvert personalities as well as positive and negative moods. Then, we present two preregistered (`aspredicted.org`) user studies examining the perception of VCs' personality (extroversion vs. introversion) and mood, (positive vs. negative) as well as how Affect Priming influences this perception. The first study examines if Movement Energy and Mood Priming (as first form of Affect Priming) affects the assessment of the VC's personality trait extroversion. The second study examines if Movement Energy and Personality Priming (as second form of Affect Priming) affects the assessment of the VC's mood. This design enables us to compare the different effect sizes of the two different forms of Affect Priming.

## 2 BACKGROUND

### 2.1 Personality, Mood, and their Influence on Motion

For this work, we rely on the OCEAN personality model that describes personality with five factors: Openness, Conscientiousness, Extroversion, Agreeableness, and Neuroticism [25]. According to this model, personality can be defined as "the relatively enduring styles of thinking, feeling, and acting that characterize an individual" [5]. The model

is readily applicable and understandable due to its coherence and the orthogonality of its traits and was already applied to create and assess static VCs [29]. In this work, we will focus on extroversion since it is the most commonly studied of those personality traits and can be recognized by humans with a high rate in VCs [8]. Extroversion and its opposite introversion is characterized by different facets, like Assertiveness, Activity, Excitement Seeking [5]. An extrovert person tends to be social, active, and dominant and has a tendency to experience positive emotions [25]. Extroversion is also reflected in motion. Extroverted people move their elbows and hands further away from the body [23]. Their gestures are fast, frequent, energetic, and broad [23, 24]. According to Laban Movement Analysis, a well established technique to systematically evaluate human motion, the time component in extrovert movements is rather sudden, which is reflected by more spacious [37] and faster movements [8]. The findings of Smith and Neff [37] illustrate that the perception of extroversion could be enhanced by the following alterations: spread fingers, increased velocity/stroke size, moving the gesture upwards.

The mood concept denotes a medium-term affect state that occurs independently of specific objects or events. It is a diffuse feeling that influences perception and cognitive processes [27]. A positive mood is linked to more optimism and confidence, and a negative mood, respectively, is related to avoidant and defensive behavior [12]. A person can be in different moods, for example, in a happy or a sad mood [36]. Mood is also reflected in motion. It influences motion aspects such as acceleration in walking [7] and waving [3], happiness causes an acceleration in both walking speed and waving motion.

Walking is particularly well suited to convey affective states because it is i.a. dependent on alterations in gait kinematics [6] and also waving can be used to communicate different emotional states [1, 3]. Therefore, we chose these two movements in our study, as both are well suited to communicate affective states such as personality and mood. Moreover, both movements have the advantage of being relatively simple which facilitates the application of movement alterations.

## 2.2 Interplay between Extroversion and Mood

Between the long-term personality trait of extroversion and the medium-term positive mood (described in this work as HAPPY), there seems to be a robust link [30]. Comparing the influence on motion, it seems that extroversion and HAPPY mood affect motion similarly. An extrovert person appears to show similar motion patterns as a HAPPY person. Therefore in this work, we will differentiate between *High Movement Energy* and *Low Movement Energy*, whereby the former is connected to extroversion and HAPPY mood and the latter to introversion and SAD mood (medium-term negative mood).

## 2.3 Affect Priming

The expectation towards and the perception of another person is influenced not only by the other person itself but also by external information about this person. This so-called contextual priming means influencing or changing a set of attitudes and globally of thinking, feeling, and acting by a particular induction [26]. Contextual information provides crucial information for the evaluation of other objects as in the real world objects and their environment have a strong relationship [39]. The idea behind priming, in general, is that a stimulus (prime) can activate previously learned cognitive structures, thereby influencing the evaluation of another stimulus [10]. In contextual priming, context means the environment in which a stimulus is perceived and includes any preceding or surrounding information [40]. Contextual priming might contain cues that influence expectation towards and the perception of another person [10]. The prime stimulus hereby affects the expectation towards and the perception of another person by increasing

the probability of activating associated attributes and biases [17, 38]. Research on priming effects often focuses on priming the affective state of the person who is doing an evaluation, who are mostly the participants themselves [34]. Using the affective priming paradigm, research investigates whether the evaluation of a priming stimulus affects the processing of subsequent stimuli [20]. One other type of contextual priming information could also be the affective state, like personality or mood, of a stimulus person that has to be assessed. In the field of social psychology, there are experiments examining priming with various personality trait adjectives. Higgins et al. (1977) found that when an ambiguous character in a story should be described, participants tended to use the primed personality concepts [18]. However, if the same applies also for a VC that is visually represented, remains unclear. Though priming was studied in the context of virtual characters [35], priming of the affective state of a VC and it's resulting effects on the perception of VCs' seem to be understudied. Therefore, this study examines the effect of Affect Priming, in particular mood (positive vs. negative) and the personality trait extroversion, on the perception of VCs' mood and extroversion.

## 2.4 Perception-driven Animation Editing

The generation of character motion that triggers the perception of specific personalities or moods has been already the subject of research for many years [4, 14]. To create the "personality-driven" video material for these studies, we followed the findings of Durupinar and colleagues [8], which identified, by the collection of the opinions of multiple subjects, the motion qualities to transform a "neutral" animation according to a given OCEAN personality profile. The advantage with respect to recording the performance of a single professional actor, or the manual work of a professional animation, is that the mapping model, despite maybe less expressive, generalizes better than the specific subjective interpretation of a single performer/artist.

## 3 VIDEO MATERIAL PREPARATION

The goal of the material preparation is to generate two VCs (one male and one female), animate them with two different motion captured data (a walking cycle and a "waving hello" gesture), and derive two Movement Energy levels (high and low) of each motion. Finally, render the eight resulting motions to video files using both a frontal and a side camera view. The material can be found on $OSF$[1].

We used Mixamo (`mixamo.com`) to generate both a female and a male character, and animate them with "neutral" *walking* and *waving* animations. We focused in choosing characters with a neutral appearance that would not influence perceived mood and personality attributes. In a post-editing phase, we used Blender (`blender.org`) to modulate the perceived personality of the initial animations. The motion provided by Mixamo comes from motion capture sessions, thus featuring high-density (30 Hz) key frames (KF), which provide natural and realistic motion, but, with respect to manual editing, leaves less room for post-production adjustments [41]. Hence, we post-edited the animations as proposed by Gokani et al. [16].

Table 1. Grouping of bones in the skeleton (L/R=left/right)

| Group | Bones |
|---|---|
| $Arm_{[L/R]}$ | Shoulder, Arm, Forearm, Wrist, Fingers. |
| $Leg_{[L/R]}$ | UpLeg, Leg, Foot, ToeBase. |
| $Spine$ | Hip, Spine, Spine1, Spine2, Neck, Head. |

[1]Experiment material available at: https://osf.io/h3k5t/?view_only=e153ab6eff8b47f48082fed29c403a29

We followed the PERFORM approach [8] and created a user interface in Blender to allow the user inputting the levels of Shape and Effort qualities that are in turn applied to the animation of the VC. The motion editing has been realized by manually applying a 5-step procedure separately for each of the bone groups listed in Table 1:

(1) Define animation *phases*;
(2) Adjust angular offset;
(3) Adjust motion amplitude;
(4) Animation curve smoothing;
(5) Time warping and time scaling.

Differently from PERFORM, which applied the motion editing using Inverse Kinematics (IK) chains, we used *Forward Kinematics*. The changes were applied separately to each of 3 color coded bone groups as shown in Fig. 2. This mixamo humanoid hierarchical skeleton structure's root is the Hip bone. Root bone is where hierarchical bone structure start to propagate as parent-child bones chains. It is important to respect the parenting order when the changes are applied in the motion curves of individual bones. For example Shoulder bone is the most parent bone in left or right Arm group and fingers are the most child bones in that group. Therefore, animation changes are first applied to shoulder bone first, in the Arm group, and fingers in the end. Such that we could preserve naturalness of the movement while preserving synchronous movement.

### 3.1 Animation Phases

The execution of a body gesture can be divided into four main segments; preparation, stroke, hold, and recovery [19]. For our case, hold is not used.

- Preparation: the phase of motion that leads from a relaxed position to the stroke phase.
- Stroke: the phase of motion where the animation dynamics of Effort and Shape are clearly expressed.
- Recovery: the phase motion that leads the stroke into a relaxed position.

The identification of the three animation phases consists of marking the two KFs defining the beginning and the end of the *stroke* phase.

Following the PERFORM terminology, the transition between phases, together with the beginning and the end of the animation, are marked as *Goal* points, which are characterised by a pause in the motion. KFs in between two *Goal* points are called *Via* points (see Fig. 3).

One of the major challenges in fully automating the animation process is bones having different goal points. For example, a shoulder bone may not have the same *Goal* points as its respective child shoulder bone. This depends on the type of the motion. Therefore, manual selection of *Preparation*, *Stroke*, *Recovery* for individual bones animations were needed.

### 3.2 Adjusting Angular Offset

In 3 dimensional space, bones motion can be defined by rate at which rotation angles $\{\theta_x, \theta_y, \theta_z\}$ change over time. Fig. 4 illustrates the right clavicle area, where applying a $\delta_x$ angular offset around *bone head* would move the all of the bone attached to *bone tip* away or closer to the body defining the trajectory of its children bones. Correspondingly, it is used to manipulate 3 dimensional bone trajectories. Durupinar et al. [8] used similar technique based on inverse kinematics and defined *Shape* qualities of VCs. *Enclosing/Spreading*, *Retreating/Advancing*, and *Sinking/Rising*, are well

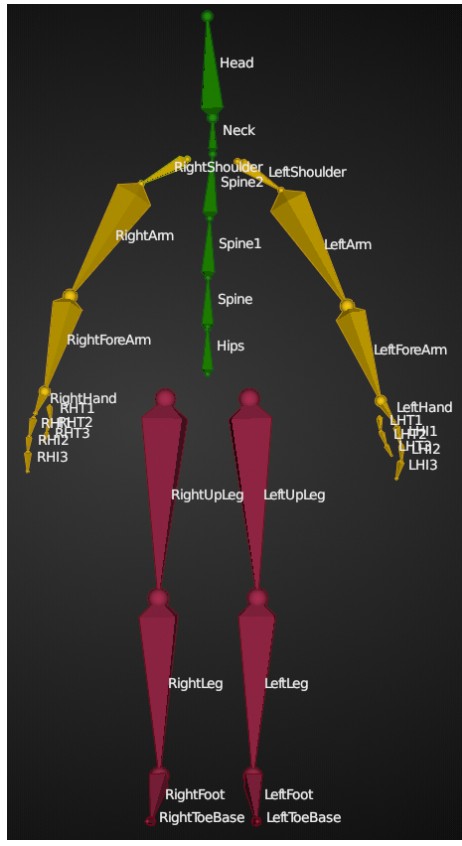

Fig. 2. The humanoid skeletal structure

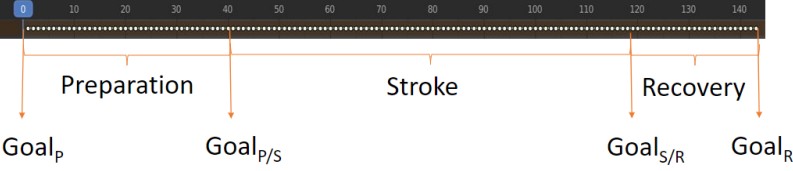

Fig. 3. Animation Timeline

described in their study [8]. In this study, we define angular offsets $\{\delta_x, \delta_y, \delta_z\}$ for every bone, which allows for a finer motion adjustment and let us take advantage of the correlation with the *Shape* qualities defined by PERFORM study.

Fig. 5a illustrates a skeleton prior to any angular offset. Fig. 5b is with angular offsets along the X axis (Blender world coordinate) to both left and right Arm bones in $Arm_{[L/R]}$ group. In Fig. 5c, we super imposed both 5a and 5b to show the effect in angular offset around -Y axis in blender world coordinates or +Z axis in blender local bone coordinates.

High and low Movement Energy were achieved in the following way.

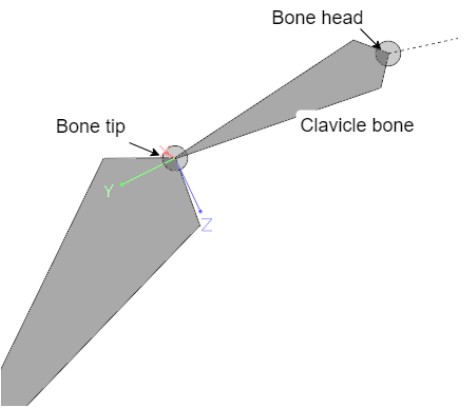

Fig. 4. Clavicle bone closeup.

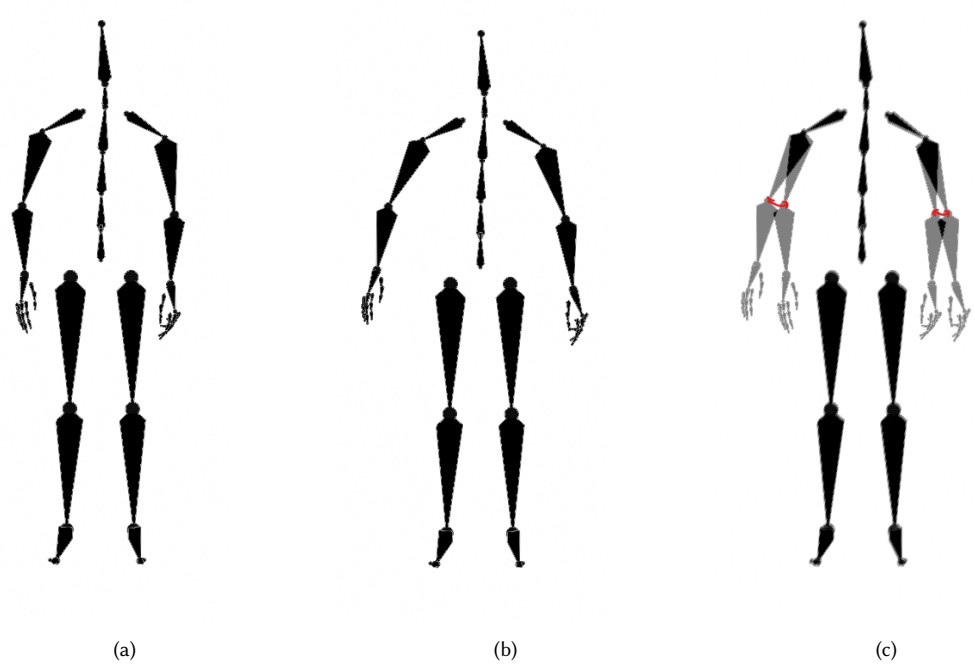

(a)                (b)                (c)

Fig. 5. (a) the standard pose, (b) after applying an angular offset to the arms, and (c) both superimposed.

High Movement Energy: Spreading was applied by moving $Arm_{[L/R]}$ and $Leg_{[L/R]}$ bones away from the centre axis of the skeleton. Retreating and Rising were applied to $Spine$ and $Arm_{[L/R]}$ groups.

Low Movement Energy: Enclosing was applied by moving $Arm_{[L/R]}$ and $Leg_{[L/R]}$ bones closer to the centre axis of the body. Advancing and Sinking were applied to $Spine$ and $Arm_{[L/R]}$ groups.

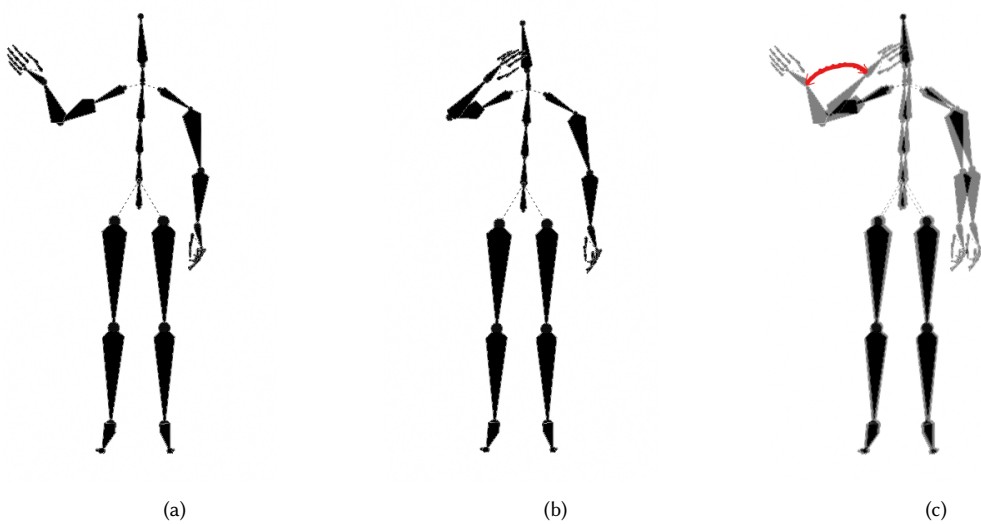

(a)       (b)       (c)

Fig. 6. Spanning space for *waving*: (a) goal frame 27, (b) goal frame 34, and (c) superimposed.

### 3.3 Adjusting Motion Amplitude

According to the literature, Extroverted people are explained as people who tend to perform gestures faster, frequent, energetic, and broader. We needed a method to control space spanned during the motion between two *Goal* points. Subsequently, control broadness of bone movements at a bone level. The red arrow in Fig. 6c shows the controlled spanned space of Arm bone moving through *Via* points. Fig. 6a and 6b shows $Goal_{P/S}$ and $Goal_{S/R}$ of the wave motion in the controlled space. Controlling of the spanning space is created by multiplying zero-meaned bone rotation values of each KF with a vector $< S_x, S_y, S_z >$, where each $S$ are positive real values. When $0 < S < 1$ in any axis, it dampens the gesture amplitude while narrowing the movement space. When $S > 1$ in any axis, then the multiplication heightens the spanned space while creating broader movements.

High Movement Energy: $S > 1$ leads to *indirect* and a *free* movements.
Low Movement Energy: $0 < S < 1$ leads to *direct* and *bounded* movements.

### 3.4 Smoothing

After applying angular offsets and modulating motion amplitudes, unwanted misalignment and sudden jumps occur in the animation curves in correspondence of the *Goal* points between motion phases. The *smoothing* is a technique to ease motion curves in order to soften those transitions.

We used Robert Penner easing method to interpolate animation curves [31]. Given a transition *Goal* point, the smoothing involves a number of neighbour *Via* points. For each smoothing, we included only two *Via* points from the stroke phase and 5-10 (depending on visual quality) *Via* points from preparation and recovery phases.

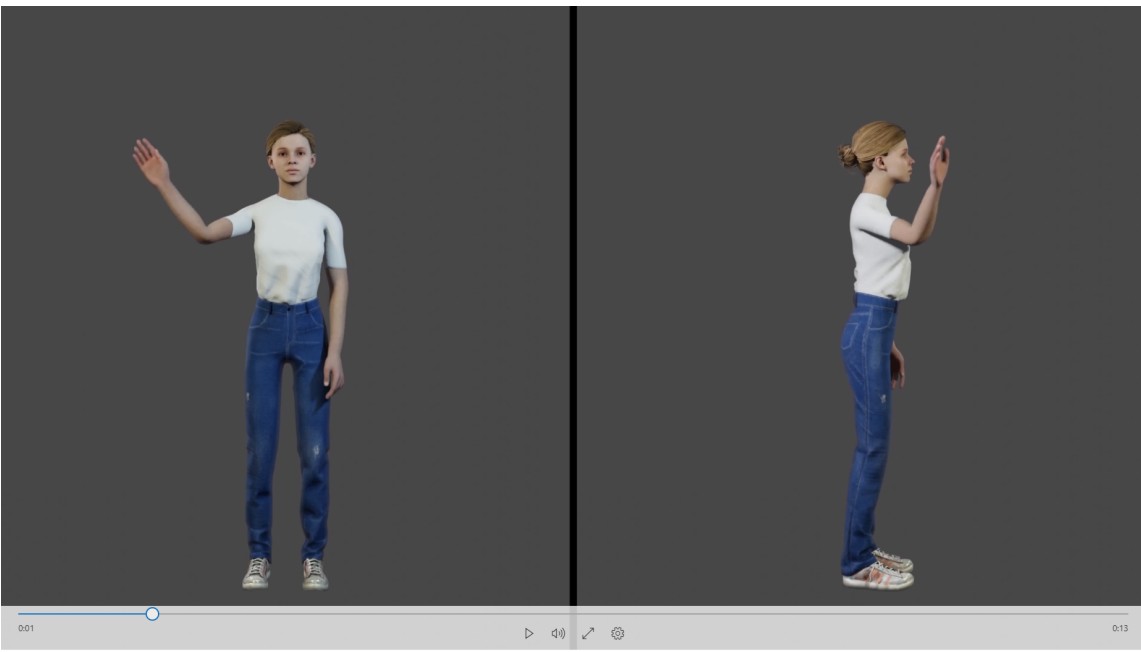

Fig. 7. Sample frames of the female character waving with a high-energy movement

### 3.5 Time Scaling and Time Warping

Time scaling modulates the overall duration of the animation through a time-scale multiplier, thus increasing or decreasing the playback speed. Differently, time warping is a transformation altering the dynamics of the curve, through acceleration and deceleration, still preserving its duration; this allows for the realization of *anticipation* and *overshoot* animation effects [8]. Time manipulation is applied over the full range of the animation, not only to the stroke.

In Blender, both time scaling and warping are implemented through the non-linear animation (NLA) editor, which gives to the user the possibility to visually edit time-related transformations through the direct manipulation of the control points of Bezier curves [11]. A proper combination of scaling and warping allows for the modulation of the speed and the dynamics of a gesture without leading to unnatural movements.

High Movement Energy: Increasing the speed of the motion leads to more *Sudden* movements.

Low Movement Energy: Decreasing the speed of the motion leads to more *Sustained* movements.

For both energy values, time warping was introduced to accelerate the *preparation* phase and to decelerate the *recovery* phase, in order to keep the motion physically believable.

After editing, the eight videos needed for the study are rendered. Fig. 1 shows an example of high and low energy versions of the male waving, while Fig. 7 shows a frame of a video as used during the studies, with both frontal and side views. The supplementary video material contains all the videos used in our user studies.

## 4  STUDY 1

This study aimed to examine if Movement Energy and Mood Priming affects the VC's personality assessment, in particular extroversion, using a 2 (Movement Energy: high vs. low) x 3 (Mood Priming: no vs. happy vs. sad) within subjects design. We included two more variables, namely 2 (Gender: male vs. female) x 2 (Movement Type: walking vs. waving) to generate a bigger variance in the stimulus material. We did not have hypotheses regarding Gender or Movement Type.

*Hypothesis 1*: Movement Energy will influence the VC's personality assessment. The VC with high Movement Energy is assessed more extrovert than the VC with low Movement Energy.

*Hypothesis 2*: Mood Priming will influence the VC's personality assessment. *2a)* The happy primed VC is assessed more extrovert than the sad primed VC. *2b)* The happy primed VC is assessed more extrovert than the not primed VC. *2c)* The not primed VC is assessed more extrovert than the sad primed VC. Overall, the pattern should be: $\text{Extroversion}_{\text{Happy Priming}} > \text{Extroversion}_{\text{No Priming}} > \text{Extroversion}_{\text{Sad Priming}}$.

*Hypothesis 3*: There is an interaction between Movement Energy and Mood Priming.

### 4.1  Methods

*Participants.* After excluding 12 participants (e.g., low scores in attention checks), the sample consisted of 125 participants mostly from Germany, Sri Lanka and India (80 female, $M_{\text{age}}$ = 27.62 years, $SD_{\text{age}}$ = 8.55 years). We based our sample size on an a priori sample planning using G*Power [9]. Participants were recruited via social networks and got the chance to take part in a lottery to win one out of five vouchers (10 Euro each) for an online store.

*Procedure.* After agreeing to the data policy, participants answered the demographic questions. To make sure personality was considered as long-term affective state, a definition was given. Afterward, participants saw three times the set of 8 videos (randomized). The first time, participants assessed the VC's personality for each video without priming. The second and third time participants assessed the VC's personality for each video with the happy or sad priming (randomized). In total, every participant rated the VC's personality 24 times which took about 25 minutes. The survey could be completed in English or German.

*Material.* For each of the three Mood Priming conditions, we had the same set of 8 *videos* (Sec. 3). In this paper, we focus on Movement Energy and Priming's effects, therefore we omit the factors gender and movement for our analysis. The videos showed the VC from a frontal view and the side (Fig. 7).

The *Mood Primings* were operationalized by giving different instructions for answering the personality questionnaire. No priming was introduced with "I see this virtual character as someone who ...", happy and sad priming were introduced with "I see this HAPPY virtual character as someone who …" and "I see this SAD virtual character as someone who …".

*Measurements.* As every participant had to assess 24 VCs (2 Movement Energy x 3 Mood Priming x 2 Gender x 2 Movement Type), we used an economic, but still psychometrically sound questionnaire. VC's *Personality* was rated with the four Extroversion items of the *BFI-K* [32] on a 5-point scale ranging from 1 (*highly disagree*) to 5 (*highly agree*). Cronbach's Alpha ranged from .70 to .92.

*Attention checks.* Five items ensured that participants attentively read the questions (e.g., "What did the virtual characters wear, jeans or shorts?"). The items included a sentence explaining their purpose. Participants with more than one incorrect answer were excluded.

*Demographics* included questions about gender, age, education level, nationality and experience with virtual characters.

### 4.2 Results

To test our hypotheses, we calculated a 2 (Movement Energy: high vs. low) x 3 (Mood Priming: no vs. happy vs. sad) repeated measures ANOVA.

*Hypothesis 1* stated that participants will assess the VC in the high Movement Energy ($M = 3.44$, $SD = 0.52$) condition more extrovert than the VC in the low Movement Energy condition ($M = 2.28$, $SD = 0.38$). We found a significant main effect of Movement Energy ($F(1,124) = 397.43$, $p < .001$, $\eta_p^2 = .76$), supporting hypothesis 1.

*Hypothesis 2* stated a main effect of the Mood Priming ($M_{no} = 2.77$, $SD_{no} = 0.39$; $M_{sad} = 2.85$, $SD_{sad} = 0.51$; $M_{happy} = 2.97$, $SD_{happy} = 0.43$), which we could find in our data (Greenhouse-Geisser corrected $F(1.53,190.24) = 9.01$, $p < .001$, $\eta_p^2 = .07$). Therefore, hypothesis 2 was confirmed by our data. Moreover, the contrasts showed that the happy primed VC was assessed more extrovert than the sad primed VC ($F(1,124) = 4.38$, $p = .019$, $\eta_p^2 = .03$), as well as the not primed VC ($F(1,124) = 36.94$, $p < .001$, $\eta_p^2 = .23$). These results support hypotheses 2a and 2b. There was no significant difference between the not primed and the sad primed condition ($F(1,124) = 2.71$, $p = .051$, $\eta_p^2 = .02$). Thus, there was no support for hypothesis 2c.

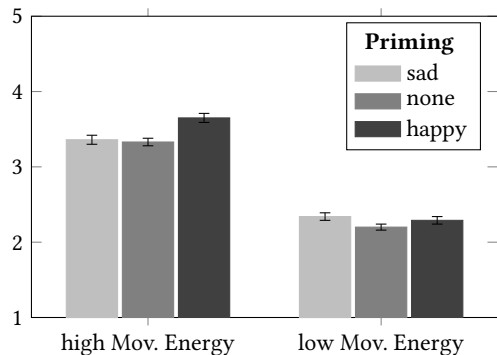

Fig. 8. $N = 125$. Personality ratings for each condition. Error bars represent standard errors; higher values represent higher extroversion.

Regarding *Hypothesis 3*, we found a significant interaction effect between the variables Movement Energy and Mood Priming ($F(2,248) = 14.59$, $p < .001$, $\eta_p^2 = .11$). For the high Movement Energy conditions the happy primed VC was assessed more extrovert than the sad primed VC ($F(1,124) = 18.68$, $p < .001$, $\eta_p^2 = .13$), as well as the not primed VC ($F(1,124) = 51.97$, $p < .001$, $\eta_p^2 = .30$). There was no significant difference between the not primed and the sad primed condition ($F(1,124) = 0.28$, $p = .598$, $\eta_p^2 = .00$) on the high Movement Energy level.

For the low Movement Energy conditions the happy primed VC was not assessed more extrovert than the sad primed VC ($F(1,124) = 0.47$, $p = .49$, $\eta_p^2 = .00$), neither the not primed VC ($F(1,124) = 3.76$, $p = .06$, $\eta_p^2 = .03$). There was a significant difference between the not primed and the sad primed condition ($F(1,124) = 5.47$, $p < .05$, $\eta_p^2 = .04$) on the low Movement Energy level.

### 4.3 Discussion Study 1

The first study examined the influence of Movement Energy and Mood Priming on the personality assessment of a VC. Our results show that VCs animated with high Movement Energy are perceived as more extroverted, which was hypothesized and goes in line with existing research [8, 23, 24]. Moreover, we found evidence that Mood Priming

affects how VCs are perceived. VCs that were primed as being happy were perceived as more extroverted than the ones without priming and sad priming. The difference between happy priming and the other two seems to be driven by these VCs presented with a high Movement Energy.

## 5 STUDY 2

This study aimed to examine if Movement Energy and Personality Priming affects the VC's mood assessment with a 2 (Movement Energy: high vs. low) x 3 (Personality Priming: no vs. extroverted vs. introverted) within subjects design. We included two more variables, namely 2 (Gender: male vs. female) x 2 (Movement Type: walking vs. waving) to generate a bigger variance in the stimulus material. We did not have hypotheses regarding Gender or Movement Type.
*Hypothesis 1*: Movement Energy will influence the VC's mood assessment. The VC with high Movement Energy is perceived being in a better mood than the VC with low Movement Energy
*Hypothesis 2*: Personality Priming will influence the VC's mood assessment. 2a) The extroverted primed VC is being in a better mood than the introverted primed VC. 2b) The extroverted primed VC is being in a better mood than the not primed VC. 2c) The not primed VC is being in a better mood than the introverted primed VC. Overall, the pattern should be: $\text{Mood}_{\text{Extrovert Priming}} > \text{Mood}_{\text{No Priming}} > \text{Mood}_{\text{Introvert Priming}}$.
*Hypothesis 3*: There is an interaction between Movement Energy and Personality Priming.

### 5.1 Methods

*Participants.* After excluding 7 participants (e.g., low scores in attention checks), the sample consisted of 157 participants mostly from Germany, Sri Lanka and Italy (95 female, $M_{\text{age}}$ = 27.37 years, $SD_{\text{age}}$ = 8.66 years). We based our sample size on an a priori sample planning using G*Power [9]. Recruiting and incentive was similar to study 1.

*Procedure.* The procedure was the same like in study 1 apart from the following: The personality definition was exchanged with a mood definition, the VC's personality was primed and mood was assessed.

*Measurements.* VC's *mood* was rated on four items adapted from the PHQ-4 health questionnaire [21]. To capture negative mood, both items of the depression scale of the PHQ-4 were adopted in a moderated form. Two reversed items were developed based on the existing. Items were introduced with "*I see this virtual character as someone who...*" and ended: "*is cheerful, joyful.*", "*has little interest and enjoyment in activities.*", "*is depressed, melancholy.*", "*is looking forward to activities.*". Items were assessed on a 5-point scale ranging from 1 (*highly disagree*) to 5 (*highly agree*). Cronbach's Alpha ranged from .65 to .87.
The same *attention check items* and *demographic questions* as in study 1 were used.

*Material.* For the three Personality Priming conditions, we created the same set of eight *videos* as in Study 1.
The *Personality Primings* were operationalized by giving different instructions for answering the mood questionnaire. No priming was introduced with "*I see this virtual character as someone who ...*", the extroverted and introverted priming was introduced with "*I see this EXTROVERTED virtual character as someone who ...*" and "*I see this INTROVERTED virtual character as someone who ...*".

### 5.2 Results

To test our hypotheses, we calculated a 2 (Movement Energy: high vs. low) x 3 (Personality Priming: no vs. extroverted vs. introverted) repeated measures ANOVA.

*Hypothesis 1* stated that participants will assess the VC in the high Movement Energy condition ($M$ = 3.48, $SD$ = 0.51) as being in a better mood than the VC in the low Movement Energy condition ($M$ = 2.61, $SD$ = 0.39). We found a significant main effect of Movement Energy ($F(1,156)$=317.19, $p$<.001, $\eta_p^2$=.67), supporting hypothesis 1.

*Hypothesis 2* stated a main effect of the Personality Priming ($M_{no}$ = 2.90, $SD_{no}$ = 0.43; $M_{extrovert}$ = 3.03, $SD_{extrovert}$ = 0.41; $M_{introvert}$ = 3.20, $SD_{introvert}$ = 0.49), which we could find in our data (Greenhouse-Geisser corrected $F(1.89,294.94)$=26.11, $p$<.001, $\eta_p^2$=.14). Therefore, hypothesis 2 was confirmed by our data. Moreover, the contrasts showed that the introverted primed VC was assessed as being in a better mood than the extroverted primed VC ($F(1,156)$ = 13.81, $p < .001$, $\eta_p^2$ = .08), as well as the not primed VC ($F(1,156)$ = 12.11, $p < .001$, $\eta_p^2$ = .07). Thus, there was no support for none of the hypotheses 2a and 2c. The extroverted primed VC was assessed significantly as being in a better mood than the not primed one ($F(1,156)$ = 52.56, $p < .001$, $\eta_p^2$ = .25) supporting hypothesis 2b.

Regarding *Hypothesis 3*, we found a significant interaction effect between the variables Movement Energy and Mood Priming ($F(2,312)$ = 16.39, $p < .001$, $\eta_p^2$ = .10). For the high Movement Energy conditions the not primed VC was assessed as being in a worse mood than the introverted primed VC ($F(1,156)$ = 42.00, $p < .001$, $\eta_p^2$ = .21), as well as the extroverted primed VC ($F(1,156)$ = 37.00, $p < .001$, $\eta_p^2$ = .19). There was no significant difference between the introverted primed and the extroverted primed condition ($F(1,156)$ = 0.70, $p = .403$, $\eta_p^2$ = .00) on the high Movement Energy level.

For the low Movement Energy condition the introverted primed VC was assessed more as being in a better mood than the not primed VC ($F(1,156)$ = 35.56, $p < .001$, $\eta_p^2$ = .19), as well as the extroverted primed VC ($F(1,156)$ = 25.46, $p < .001$, $\eta_p^2$ = .14). There was no significant difference between the not primed and the extroverted condition ($F(1,156)$ = 0.00, $p = .981$, $\eta_p^2$ = .00) on the low Movement Energy level.

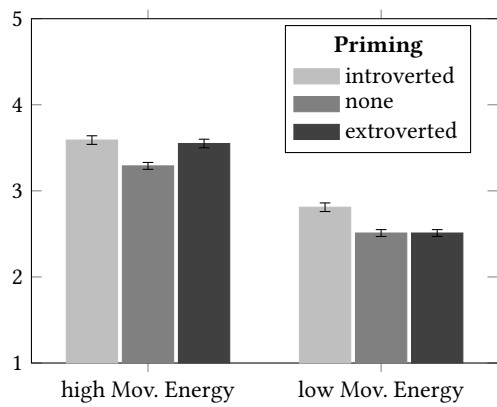

Fig. 9. *N* = 157. Mood ratings for each condition. Error bars represent standard errors; higher values represent better mood.

### 5.3 Discussion Study 2

The second study aimed to examine the influence of Movement Energy and Personality Priming on VCs' mood assessment. Our results show that VCs animated with high Movement Energy are perceived as being in a better mood which was hypothesized and goes in line with existing research [7]. Moreover, we found evidence that Personality Priming affects how VCs are perceived. However, this priming effect had another direction as expected. The introverted primed VC was perceived as being in the best mood. This applies especially for the VCs with low Movement Energy.

This means that from the VCs showing a low Movement Energy, these ones are perceived as being in the best mood that were described as being introverted. One explanation for this could be the congruence of both stimuli. The animations of low Movement Energy are based on human motion that is perceived as rather introverted and as being in a bad mood. If a VC shows this motion and is described as introverted, both sources of information are in line, which might lead to the perception of a VC in a good mood.

## 6 DISCUSSION, LIMITATIONS AND FUTURE WORK

The goal of the two studies presented in this paper was to examine how Movement Energy and contextual priming of affective information influence VCs' perception. In both studies, we could show with extremely high effect sizes that Movement Energy affects the perception of personality and mood of VCs. VCs animated with a high Movement Energy reflected in wider, more straightened, and sudden movements seem to be perceived more extroverted and in a better mood. Regarding contextual priming, we found that Personality Priming has higher effects on mood perception than Mood Priming on personality perception. It might be that Personality Priming has more prominent effects because personality is considered as a long-term affective state and influences many aspects of a person [5, 25]. On the other side, mood is a medium-term affective state and is rather diffuse [27]. Therefore, also its influence on the perception of personality might be smaller. Regarding the Mood Priming, as expected, the happy primed VCs were assessed the most extrovert. This applies especially for these VCs showing a high Movement Energy, where both stimuli, the actual movement of the VC and the prime are congruent. However, for these VCs showing a low Movement Energy, the happy primed VCs are not perceived differently regarding their personality compared to the sad or not primed ones. Regarding the Personality Priming, against our assumption, the introverted primed VCs were perceived as being in the best mood. This result still applies when analyzing the two movements walking and waving, separately. Especially, the VC expressing low Movement Energy and being described to be introverted, was perceived in the best mood. A reason for this result might be congruence of both stimuli as the VC that is described as introverted is showing a behaviour that matches this personality. However, why the VC in the opposite condition, the one expressing high Movement Energy and being described to be extroverted is not perceived happier than the one described as introverted, remains unclear. Future work should examine the effect of Personality Priming further.

Like in every study, there are several limitations. On the technical side, the options in selecting neutral characters and animations from a library are limited. Furthermore, animating our own character with presorted animations (retargeting) often leads to distorted, unnatural character motions. In our future studies, we wish to work with a motion capture setup and character development software to improve our animation pipeline. Regarding the study, we focused only on the personality factor extroversion. As also other other personality factors might be reflected in movement, future studies should look into this. Moreover, we created videos showing VCs moving based on human motion patters. We did not compare this animations against videos showing real humans performing the same movements. Using a motion capture setup, it would be possible to compare both. However, we found in our study similar effects like in studies examining human motions. This can be seen as evidence that VCs are perceived similarly to humans.

As already mentioned, objects and their environment are strongly connected [39]. In this study, we manipulated Movement Energy and Affect Priming. The perception of VCs, however, can be influenced by many other VC determined factors, like clothing, facial expression and facial features, or contexts in which the VC is presented. Moreover, also determinants of the person who assesses the VC can affect the perception. Therefore, future research should investigate these factors.

## 7 SUMMARY

The aim of the two studies we conducted was to examine the impact of specific movement alterations reflected in different Movement Energies as well as the effect of Affect Priming on the perceived personality and mood. The first study examined the perception of personality whilst the second study examined mood with the same videos. The results point to high Movement Energy being perceived as more extrovert and happy compared to low Movement Energy. The priming of mood and personality showed a strong influence on the ratings in both studies. How users perceive VCs is crucial for several applications, for example, social training systems in which VCs overtake the role of interaction partners and enable difficult social situations to be experienced virtually [2, 15].

## ACKNOWLEDGMENTS

This work is partially funded by the EU Horizon 2020 program within the MindBot project (Grant agreement ID: 847926).

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
