# OpenReview forum: "Influence of Movement Energy and Affect Priming on the Perception of Virtual Characters Extroversion and Mood"
_ACM.org/ICMI/2021/Workshop/GENEA — GENEA Workshop 2021 Oral_

### Official Review · Reviewer_TCvC · 2021-07-13
**This paper evaluated the influence of movement energy (size and speed) and affect priming on the perception of virtual character personality (extroverted to introverted) and mood (happy to sad), in two events (hand shaking and walking). The evaluation results are partly interesting, partly obvious, and partly not-explainable. But overall, the experiment design of considering affect priming was interesting.**

**Rating:** 6
**Confidence:** 4

**Review:**

Overall results first indicated that higher energy are perceived as more extrovert and happier, which are not surprising results.

The influences of priming indicated slightly higher scores for congruent factors (higher energy and happy priming resulting in higher extroversion scores, and lower energy and introverted priming resulting in higher mood scores.) The results for introverted priming was interesting.

However, why the "without priming" condition resulted in slightly lower mood scores in comparison to the "with priming" conditions in high movement energy is still an open question.

Some points need clarification:

- How about the effects of facial expressions? Was the face kept neutral in happy and sad moods?

- What is the reason forward kinematics are used instead of inverse kinematics?

- In section 3.3, what are "indirect" and "free" movements?

The paper should be accompanied by a video. It is questionable if the Virtual Character motion quality did not affect the subjective scores of different priming conditions.

---

### Official Review · Reviewer_Riyn · 2021-07-16
**Clear Accept**

**Rating:** 8
**Confidence:** 4

**Review:**

Summary:
In this paper, the authors explore the relationship between the animation of a virtual character's nonverbal behaviors, and how it is perceived. The paper reports on two studies investigating the effects of "movement energy" (broader, quicker, more sudden movements), and "affect priming" (framing either the character's mood or personality) on how the character is perceived (mood or personality). In Study 1, they found that virtual characters were more likely to be perceived as extroverted if they were animated with high movement energy and were framed as being "happy". In Study 2, they found partially symmetric results, in that virtual characters animated with high movement energy were perceived to be in a happier mood, but being framed as "extroverted" did not necessarily lead to perceptions of being in a better mood. Only the characters framed as "introverted" and animated congruously with low movement energy were perceived as being happier.

Strengths
- The studies and analyses are sound and thoroughly described. The authors do not overextend themselves in deriving any unsupported conclusions.
- I appreciate the level of detail about the animation pipeline behind the generation of all the characters' movements. This makes it easier for others to replicate and extend this work.
- The background does a very good job explaining how personality and mood are being operationalized in this work, and how they are related to motion.

Weaknesses:
- I'm a bit concerned that participants might have recognized seeing the same video three times during the course of the study, even though they were presented in random order. This could have led to confusion when the priming changed on each viewing. Was any such confusion observed? Were there any manipulation checks in these studies?
- At a high level, the paper could benefit from a bit more motivation on why there is a need to study *these* particular dimensions together, and not separately. The connection between movement and perceived mood and personality is clear, but the inclusion of the priming/framing variable was less motivated in the text. Why add this variable, and not others? E.g., the ones listed at the end of the paper (clothing, facial expression, facial features, other contextual factors, etc.)?

---

### Official Review · Reviewer_fk8y · 2021-07-18
**Priming results make an interesting contribution.**

**Rating:** 7
**Confidence:** 4

**Review:**

The paper describes edits that are performed on two different animation sequences to create high and low level animations of waving and walking.  These animations are then used in two perceptual studies to evaluate how the energy level of the motion and the priming impacts perceptions of agent personality and mood.

I think the material related to priming makes a novel and interesting contribution.  It is probably sufficiently established at this point that that high energy motion will increase the perception of extraversion, but there is far less work on the impacts of priming.  I found the analysis of the two different types of primes on two different measures particularly interesting.  There are also nice aspects of the study design (power analysis, appropriate sample size, etc.)

The animation approach is not clearly novel.  Amplitude edits, for instance, were proposed quite a while before [1] and the structure of the system is based on PERFORM.  If there are novel aspects, these should be highlighted, but it is not necessary for the contribution and it would also be possible to shorten this portion of the paper. There was certainly quite a bit of work in putting all of this together.

The edits seem reasonable for the proposed wave motion.  It is not clear how the walk animation was manipulated.  Was only the upper body changed?  Similar edits to the lower body could lead to foot skating.

The paper mentions supplementary videos, but the only file included with the submission is a pdf.  If a video upload option was not included, a link to a YouTube repository of the video, or similar, should be provided.  I am uncomfortable evaluating a paper that is creating and evaluating animation without seeing a video of the work, although the adjustments as described should fulfull their intended purpose.

It seems reasonable to me that the character primed to be introverted would be seen as in a better mood.  The introvert priming sets an expectation of low energy, so if the motion exceeds it (high energy) or meets it (low energy), I would expect that to be viewed more positively than for an extrovert, where high energy was expected.  The curious aspect to me is that the high energy motion with extrovert priming was also viewed as more positive than neutral.  Why do you think this is?  Is it again simply a consistent signal (extrovert sets up an expectation of high energy, and so when people see that, it reinforces the assumption and they decide the person is more positive because they have that association with extroverts)?  It would be helpful to add more discussion of this.

I did not understand the issue with goal points and automation at lines 381-385.

1.  Neff, M. and Fiume, E., 2003, July. Aesthetic edits for character animation. In Proceedings of the 2003 ACM SIGGRAPH/Eurographics symposium on Computer animation (pp. 239-244).

---

### Decision · Program_Chairs · 2021-07-19

Accept (Oral)